# Mean-field theory of graph neural networks in graph partitioning

**Tatsuro Kawamoto,  Masashi Tsubaki**
Artificial Intelligence Research Center,
National Institute of Advanced Industrial Science and Technology,
2-3-26 Aomi, Koto-ku, Tokyo, Japan
`{kawamoto.tatsuro, tsubaki.masashi}@aist.go.jp`

**Tomoyuki Obuchi**
Department of Mathematical and Computing Science, Tokyo Institute of Technology,
2-12-1 Ookayama Meguro-ku Tokyo, Japan
`obuchi@c.titech.ac.jp`

## Abstract

A theoretical performance analysis of the graph neural network (GNN) is presented. For classification tasks, the neural network approach has the advantage in terms of flexibility that it can be employed in a data-driven manner, whereas Bayesian inference requires the assumption of a specific model. A fundamental question is then whether GNN has a high accuracy in addition to this flexibility. Moreover, whether the achieved performance is predominately a result of the backpropagation or the architecture itself is a matter of considerable interest. To gain a better insight into these questions, a mean-field theory of a minimal GNN architecture is developed for the graph partitioning problem. This demonstrates a good agreement with numerical experiments.

## 1   Introduction

Deep neural networks have been subject to significant attention concerning many tasks in machine learning, and a plethora of models and algorithms have been proposed in recent years. The application of the neural network approach to problems on graphs is no exception and is being actively studied, with applications including social networks and chemical compounds [1, 2]. A neural network model on graphs is termed a graph neural network (GNN) [3]. While excellent performances of GNNs have been reported in the literature, many of these results rely on experimental studies, and seem to be based on the blind belief that the nonlinear nature of GNNs leads to such strong performances. However, when a deep neural network outperforms other methods, the factors that are really essential should be clarified: Is this thanks to the learning of model parameters, e.g., through the backpropagation [4], or rather the architecture of the model itself? Is the choice of the architecture predominantly crucial, or would even a simple choice perform sufficiently well? Moreover, does the GNN generically outperform other methods?

To obtain a better understanding of these questions, not only is empirical knowledge based on benchmark tests required, but also theoretical insights. To this end, we develop a mean-field theory of GNN, focusing on a problem of graph partitioning. The problem concerns a GNN with random model parameters, i.e., an untrained GNN. If the architecture of the GNN itself is essential, then the performance of the untrained GNN should already be effective. On the other hand, if the fine-tuning of the model parameters via learning is crucial, then the result for the untrained GNN is again useful to observe the extent to which the performance is improved.

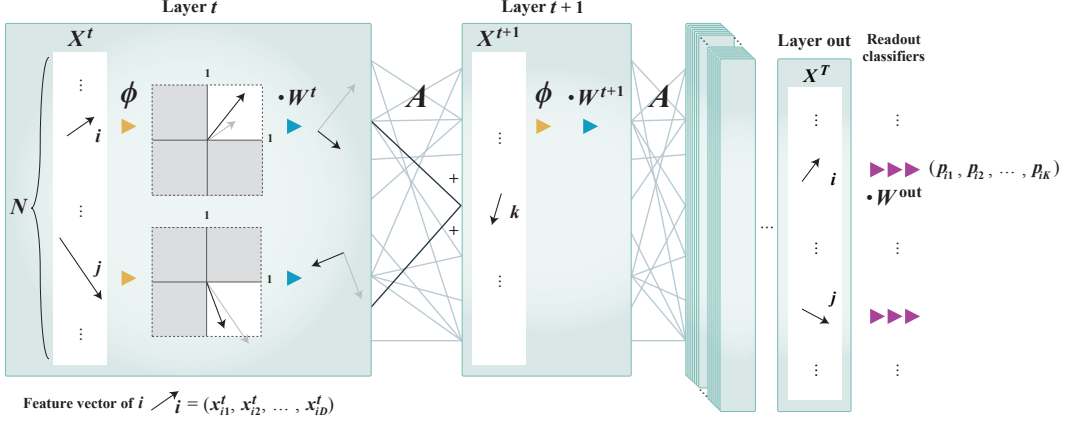

Figure 1: Architecture of the GNN considered in this paper.

Table 1: Comparison of various methods under the framework of Eq. (2).

| algorithm | domain | $\boldsymbol{M}$ | $\phi(x)$ | $\varphi(x)$ | $\boldsymbol{W}^t$ | $\boldsymbol{b}^t$ | $\{\boldsymbol{W}^t, \boldsymbol{b}^t\}$ update |
|---|---|---|---|---|---|---|---|
| untrained GNN | $V$ | $\boldsymbol{A}$ | $\tanh$ | $I(x)$ | random | omitted | not trained |
| trained GNN [5] | $V$ | $\boldsymbol{I}-\boldsymbol{L}$ | ReLu | $I(x)$ | trained | omitted | trained via backprop. |
| Spectral method | $V$ | $\boldsymbol{L}$ | $I(x)$ | $I(x)$ | QR | / | updated at each layer |
| EM + BP | $\vec{E}$ | $\boldsymbol{B}$ | softmax | $\log(x)$ | learned | learned | learned via M-step |

For a given graph $G = (V, E)$, where $V$ is the set of vertices and $E$ is the set of (undirected) edges, the graph partitioning problem involves assigning one out of $K$ group labels to each vertex. Throughout this paper, we restrict ourselves to the case of two groups ($K = 2$). The problem setting for graph partitioning is relatively simple compared with other GNN applications. Thus, it is suitable as a baseline for more complicated problems. There are two types of graph partitioning problem: One is to find the best partition for a given graph under a certain objective function. The other is to assume that a graph is generated by a statistical model, and infer the planted (i.e., preassigned) group labels of the generative model. Herein, we consider the latter problem.

Before moving on to the mean-field theory, we first clarify the algorithmic relationship between GNN and other methods of graph partitioning.

## 2 Graph neural network and its relationship to other methods

The goal of this paper is to examine the graph partitioning performance using a minimal GNN architecture. To this end, we consider a GNN with the following feedforward dynamics. Each vertex is characterized by a $D$-dimensional feature vector whose elements are $x_{i\mu}$ ($i \in V, \mu \in \{1, \ldots, D\}$), and the state matrix $\boldsymbol{X} = [x_{i\mu}]$ obeys

$$x_{i\mu}^{t+1} = \sum_{j\nu} A_{ij} \phi\left(x_{j\nu}^t\right) W_{\nu\mu}^t + b_{i\mu}^t. \tag{1}$$

Throughout this paper, the layers of the GNN are indexed by $t \in \{0, \ldots, T\}$. Furthermore, $\phi(x)$ is a nonlinear activation function, $\boldsymbol{b}^t = [b_{i\mu}^t]$ is a bias term[1], and $\boldsymbol{W}^t = [W_{\nu\mu}^t]$ is a linear transform that operates on the feature space. To infer the group assignments, a $D \times K$ matrix $\boldsymbol{W}^{\mathrm{out}}$ of the readout classifier is applied at the end of the last layer. Although there is no restriction on $\phi$ in our mean-field theory, we adopt $\phi = \tanh$ as a specific choice in the experiment below. As there is no detailed attribute on each vertex, the initial state $\boldsymbol{X}^0$ is set to be uniformly random, and the adjacency matrix $\boldsymbol{A}$ is the only input in the present case. For the graphs with vertex attributes, deep neural networks that utilize such information [6, 7, 8] are also proposed.

The set of bias terms $\{\boldsymbol{b}^t\}$, the linear transforms $\{\boldsymbol{W}^t\}$ in the intermediate layers, and $\boldsymbol{W}^{\mathrm{out}}$ for the readout classifier are initially set to be random. These are updated through the backpropagation using the training set. In the semi-supervised setting, (real-world) graphs in which only a portion of the vertices are correctly labeled are employed as a training set. On the other hand, in the case of unsupervised learning, graph instances of a statistical model can be employed as the training set.

The GNN architecture described above can be thought of as a special case of the following more general form:

$$x_{i\mu}^{t+1} = \sum_j M_{ij}\, \varphi \left( \sum_\nu \phi(x_{j\nu}^t) W_{\nu\mu}^t \right) + b_{i\mu}^t, \tag{2}$$

where $\varphi(x)$ is another activation function. With different choices for the matrix and activation functions shown in Table 1, various algorithm can be obtained. Equation (1) is recovered by setting $M_{ij} = A_{ij}$ and $\varphi(x) = I(x)$ (where $I(x)$ is the identity function), while [5] employed a Laplacian-like matrix $\boldsymbol{M} = \boldsymbol{I} - \boldsymbol{L} = \boldsymbol{D}^{-1/2}\boldsymbol{A}\boldsymbol{D}^{-1/2}$, where $\boldsymbol{D}^{-1/2} \equiv \mathrm{diag}(\sqrt{d_1}, \ldots, \sqrt{d_N})$ ($d_i$ is the degree of vertex $i$) and $\boldsymbol{L}$ is called the normalized Laplacian [9].

The spectral method using the power iteration is obtained in the limit where $\phi(x)$ and $\varphi(x)$ are linear and $\boldsymbol{b}^t$ is absent.[2] For the simultaneous power iteration that extracts the leading $K$ eigenvectors, the state matrix $\boldsymbol{X}^0$ is set as an $N \times K$ matrix whose column vectors are mutually orthogonal. While the normalized Laplacian $\boldsymbol{L}$ is commonly adopted for $\boldsymbol{M}$, there are several other choices [9, 10, 11].[3] At each iteration, the orthogonality condition is maintained via QR decomposition [12], i.e., for $\boldsymbol{Z}^t := \boldsymbol{M}\boldsymbol{X}^t$, $\boldsymbol{X}^{t+1} = \boldsymbol{Z}^t \boldsymbol{R}_t^{-1}$, where $\boldsymbol{R}_t^{-1}$ acts as $\boldsymbol{W}^t$. $\boldsymbol{R}_t$ is a $D \times D$ upper triangular matrix that is obtained by the QR decomposition of $\boldsymbol{Z}^t$. Therefore, rather than training $\boldsymbol{W}^t$, it is determined at each layer based on the current state.

The belief propagation (BP) algorithm (also called the message passing algorithm) in Bayesian inference also falls under the framework of Eq. (2). While the domain of the state consists of the vertices $(i, j \in V)$ for GNNs, this algorithm deals with the directed edges $i \to j \in \vec{E}$, where $\vec{E}$ is obtained by putting directions to every undirected edge. In this case, the state $x_{\sigma, i \to j}^t$ represents the logarithm of the marginal probability that vertex $i$ belongs to the group $\sigma$ with the missing information of vertex $j$ at the $t$th iteration. With the choice of matrix and activation functions shown in Table 1 (EM+BP), Eq. (2) becomes exactly the update equation of the BP algorithm[4] [13]. The matrix $\boldsymbol{M} = \boldsymbol{B} = [B_{j \to k, i \to j}]$ is the so-called non-backtracking matrix [14], and the softmax function represents the normalization of the state $x_{\sigma, i \to j}^t$.

The BP algorithm requires the model parameters $\boldsymbol{W}^t$ and $\boldsymbol{b}^t$ as inputs. For example, when the expectation-maximization (EM) algorithm is considered, the BP algorithm comprises half (the E-step) of the algorithm. The parameter learning of the model is conducted in the other half (the M-step), which can be performed analytically using the current result of the BP algorithm. Here, $\boldsymbol{W}^t$ and $\boldsymbol{b}^t$ are the estimates of the so-called density matrix (or affinity matrix) and the external field resulting from messages from non-edges [13], respectively, and are common for every $t$. Therefore, the differences between the EM algorithm and GNN are summarized as follows. While there is an analogy between the inference procedures, in the EM algorithm the parameter learning of the model is conducted analytically, at the expense of the restrictions of the assumed statistical model. On the other hand, in GNNs the learning is conducted numerically in a data-driven manner [15], for example by backpropagation. While we will shed light on the detailed correspondence in the case of graph partitioning here, the relationship between GNN and BP is also mentioned in [16, 17].

## 3 Mean-field theory of the detectability limit

### 3.1 Stochastic block model and its detectability limit

We analyze the performance of an untrained GNN on the stochastic block model (SBM). This is a random graph model with a planted group structure, and is commonly employed as the generative

model of an inference-based graph clustering algorithm [13, 18, 19, 20]. The SBM is defined as follows. We let $|V| = N$, and each of the vertices has a preassigned group label $\sigma \in \{1, \ldots, K\}$, i.e., $V = \cup_\sigma V_\sigma$. We define $V_\sigma$ as the set of vertices in a group $\sigma$, $\gamma_\sigma \equiv |V_\sigma|/N$, and $\sigma_i$ represents the planted group assignment of vertex $i$. For each pair of vertices $i \in V_\sigma$ and $j \in V_{\sigma'}$, an edge is generated with probability $\rho_{\sigma\sigma'}$, which is an element of the density matrix. Throughout this paper, we assume that $\rho_{\sigma\sigma'} = O(N^{-1})$, so that the resulting graph has a constant average degree, or in other words the graph is sparse. We denote the average degree by $c$. Therefore, the adjacency matrix $\boldsymbol{A} = [A_{ij}]$ of the SBM is generated with probability

$$p(\boldsymbol{A}) = \prod_{i<j} \rho_{\sigma_i\sigma_j}^{A_{ij}} \left(1 - \rho_{\sigma_i\sigma_j}^{A_{ij}}\right)^{1-A_{ij}}. \tag{3}$$

When $\rho_{\sigma\sigma'}$ is the same for all pairs of $\sigma$ and $\sigma'$, the SBM is nothing but the Erdős-Rényi random graph. Clearly, in this case no algorithm can infer the planted group assignments better than random chance. Interestingly, even when $\rho_{\sigma\sigma'}$ is not constant there exists a limit for the strength of the group structure, below which the planted group assignments cannot be inferred better than chance. This is called the detectability limit. Consider the SBM consisting of two equally-sized groups that is parametrized as $\rho_{\sigma\sigma'} = \rho_{\mathrm{in}}$ for $\sigma = \sigma'$ and $\rho_{\sigma\sigma'} = \rho_{\mathrm{out}}$ for $\sigma \neq \sigma'$, which is often referred to as the symmetric SBM. In this case, it is rigorously known [21, 22, 23] that detection is impossible by any algorithm for the SBM with a group structure weaker than

$$\epsilon = (\sqrt{c} - 1)/(\sqrt{c} + 1), \tag{4}$$

where $\epsilon \equiv \rho_{\mathrm{out}}/\rho_{\mathrm{in}}$. However, it is important to note that this is the information-theoretic limit, and the achievable limit for a specific algorithm may not coincide with Eq. (4). For this reason, we investigate the *algorithmic detectability limit* of the GNN here.

## 3.2 Dynamical mean-field theory

In an untrained GNN, each element of the matrix $\boldsymbol{W}^t$ is randomly determined according to the Gaussian distribution at each $t$, i.e., $W_{\nu\mu}^t \sim \mathcal{N}(0, 1/D)$. We assume that the feature dimension $D$ is sufficiently large, but $D/N \ll 1$. Let us consider a state $\mathsf{x}_{\sigma\mu}^t$ that represents the average state within a group, i.e., $\mathsf{x}_{\sigma\mu}^t \equiv (\gamma_\sigma N)^{-1} \sum_{i \in V_\sigma} x_{i\mu}^t$. The probability distribution that $\mathbf{x}^t = [\mathsf{x}_{\sigma\mu}^t]$ is expressed as

$$P(\mathbf{x}^{t+1}) = \left\langle \prod_{\sigma\mu} \delta \left( \mathsf{x}_{\sigma\mu}^{t+1} - \frac{1}{\gamma_\sigma N} \sum_{i \in V_\sigma} \sum_{j\nu} A_{ij} \phi(x_{j\nu}^t) W_{\nu\mu}^t \right) \right\rangle_{A, W^t, X^t}, \tag{5}$$

where $\langle \ldots \rangle_{A, W^t, X^t}$ denotes the average over the graph $\boldsymbol{A}$, the random linear transform $\boldsymbol{W}^t$, and the state $\boldsymbol{X}^t$ of the previous layer. Using the Fourier representation, the normalization condition of Eq. (5) is expressed as

$$1 = \int \mathcal{D}\hat{\mathbf{x}}^{t+1} \mathcal{D}\mathbf{x}^{t+1} \mathrm{e}^{-\mathcal{L}_0} \left\langle \mathrm{e}^{\mathcal{L}_1} \right\rangle_{A, W^t, X^t}, \qquad \begin{cases} \mathcal{L}_0 = \sum_{\sigma\mu} \gamma_\sigma \hat{\mathsf{x}}_{\sigma\mu}^{t+1} \mathsf{x}_{\sigma\mu}^{t+1} \\ \mathcal{L}_1 = \frac{1}{N} \sum_{\mu\nu} \sum_{ij} A_{ij} W_{\nu\mu}^t \hat{\mathsf{x}}_{\sigma_i\mu}^{t+1} \phi(x_{j\nu}^t). \end{cases} \tag{6}$$

where $\hat{\mathsf{x}}_{\sigma\mu}^{t+1}$ is an auxiliary variable that is conjugate to $\mathsf{x}_{\sigma\mu}^{t+1}$, and $\mathcal{D}\hat{\mathbf{x}}^{t+1} \mathcal{D}\mathbf{x}^{t+1} \equiv \prod_{\sigma\mu} (\gamma_\sigma d\hat{\mathsf{x}}_{\sigma\mu}^{t+1} d\mathsf{x}_{\sigma\mu}^{t+1}/2\pi i)$.

After taking the average of the symmetric SBM over $\boldsymbol{A}$ as well as the average over $\boldsymbol{W}^t$ in the stationary limit with respect to $t$, the following self-consistent equation is obtained with respect to the covariance matrix $\boldsymbol{C} = [C_{\sigma\sigma'}]$ of $\mathbf{x} = \mathbf{x}^t$:

$$C_{\sigma\sigma'} = \frac{1}{\gamma_\sigma \gamma_{\sigma'}} \sum_{\tilde{\sigma}\tilde{\sigma}'} B_{\sigma\tilde{\sigma}} B_{\sigma'\tilde{\sigma}'} \int \frac{d\mathbf{x}\, \mathrm{e}^{-\frac{1}{2}\mathbf{x}^\top \boldsymbol{C}^{-1} \mathbf{x}}}{(2\pi)^{\frac{N}{2}} \sqrt{\det \boldsymbol{C}}} \phi(\mathsf{x}_{\tilde{\sigma}}) \phi(\mathsf{x}_{\tilde{\sigma}'}), \tag{7}$$

where $B_{\sigma\sigma'} \equiv N\gamma_\sigma \rho_{\sigma\sigma'} \gamma_{\sigma'}$. The detailed derivation can be found in the supplemental material. The reader may notice that the above expression resembles the recursive equations in [24, 25]. However, it should be noted that Eq. (7) is not obtained as an exact closed equation. The derivation relies

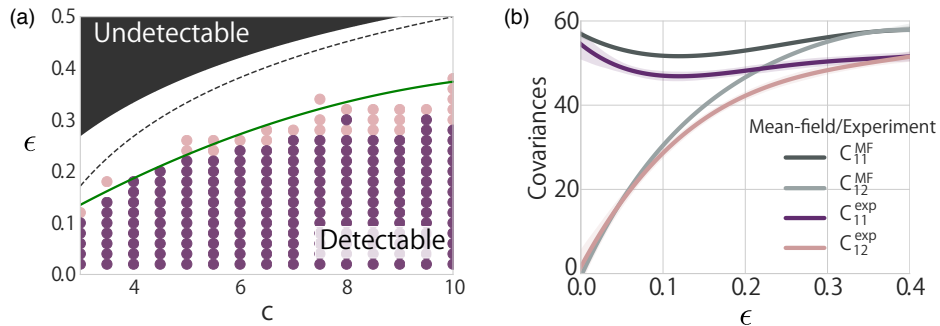

Figure 2: Performance of the untrained GNN evaluated by the behavior of the covariance matrix $C$. (a) Detectability phase diagram of the SBM. The solid line represents the mean-field estimate by Eq. (7), the dashed line represents the phase boundary of the spectral method (see Section 5 for details), and the dark shaded region represents the region above Eq. (4). The region containing points is that where the covariance gap $C_{11} - C_{12}$ is significantly larger than the gaps in the information-theoretically undetectable region. (b) The curves of the covariances $C_{11}$ and $C_{12}$ of the SBMs with $c = 8$: The mean-field estimates (gray lines that have larger values) and curves obtained by the regression of the experimental data (purple lines with smaller values).

mainly on the assumption that the macroscopic random variable $\mathbf{x}^t$ dominates the behavior of the state $\boldsymbol{X}^t$. It is numerically confirmed that this assumption appears plausible. This type of analysis is called dynamical mean-field theory (or the Martin-Siggia-Rose formalism) [26, 27, 28, 29].

When the correlation within a group $C_{\sigma\sigma}$ is equal to the correlation between groups $C_{\sigma\sigma'}$ ($\sigma \neq \sigma'$), the GNN is deemed to have reached the detectability limit. Beyond the detectability limit, Eq. (7) is no longer a two-component equation, but is reduced to an equation with respect to the variance of one indistinguishable group.

The accuracy of our mean-field estimate is examined in Fig. 2, using the covariance matrix $C$ that we obtained directly from the numerical experiment. In the detectability phase diagram in Fig. 2a, the region that has a covariance gap $C_{11} - C_{12} > \mu_{\mathrm{g}} + 2\sigma_{\mathrm{g}}$ for at least one graph instance among 30 samples is indicated by the dark purple points, and the region with $C_{11} - C_{12} > \mu_{\mathrm{g}} + \sigma_{\mathrm{g}}$ is indicated by the pale purple points, where $\mu_{\mathrm{g}}$ and $\sigma_{\mathrm{g}}$ are the mean and standard deviation, respectively, of the covariance gap in the information-theoretically undetectable region. In Fig. 2b, the elements of the covariance matrix are compared for the SBM with $c = 8$. The consistency of our mean-field estimate is examined for a specific implementation in Section 5.

## 4  Normalized mutual information error function

The previous section dealt with the feedforward process of an untrained GNN. By employing a classifier such as the k-means method [30], $\boldsymbol{W}^{\mathrm{out}}$ is not required, and the inference of the SBM can be performed without any training procedure. To investigate whether the training significantly improves the performance, the algorithm that updates the matrices $\{\boldsymbol{W}^t\}$ and $\boldsymbol{W}^{\mathrm{out}}$ must be specified.

The cross-entropy error function is commonly employed to train $\boldsymbol{W}^{\mathrm{out}}$ for a classification task. However, this error function unfortunately cannot be directly applied to the present case. Note that the planted group assignment of SBM is invariant under a global permutation of the group labels. In other words, as long as the set of vertices in the same group share the same label, the label itself can be anything. This is called the identifiability problem [31]. The cross-entropy error function is not invariant under a global label permutation, and thus the classifier cannot be trained unless the degrees of freedom are constrained. A possible brute force approach is to explicitly evaluate all the permutations in the error function [32], although this obviously results in a considerable computational burden unless the number of groups is very small. Note also that semi-supervised clustering does not suffer from the identifiability issue, because the permutation symmetry is explicitly broken.

Here, we instead propose the use of the normalized mutual information (NMI) as an error function for the readout classifier. The NMI is a comparison measure of two group assignments, which naturally eliminates the permutation degrees of freedom. Let $\boldsymbol{\sigma} = \{\sigma_i = \sigma | i \in V_\sigma\}$ be the labels of the planted group assignments, and $\hat{\boldsymbol{\sigma}} = \{\sigma_i = \hat{\sigma} | i \in V_{\hat{\sigma}}\}$ be the labels of the estimated group assignments. First, the (unnormalized) mutual information is defined as

$$I(\boldsymbol{\sigma}, \hat{\boldsymbol{\sigma}}) = \sum_{\sigma=1}^{K} \sum_{\hat{\sigma}=1}^{K} P_{\sigma\hat{\sigma}} \log \frac{P_{\sigma\hat{\sigma}}}{P_\sigma P_{\hat{\sigma}}}, \tag{8}$$

where the joint probability $P_{\sigma\hat{\sigma}}$ is the fraction of vertices that belong to the group $\sigma$ in the planted assignment and the group $\hat{\sigma}$ in the estimated assignment. Furthermore, $P_\sigma$ and $P_{\hat{\sigma}}$ are the marginals of $P_{\sigma\hat{\sigma}}$, and we let $H(\boldsymbol{\sigma})$ and $H(\hat{\boldsymbol{\sigma}})$ be the corresponding entropies. The NMI is defined by

$$\text{NMI}(\boldsymbol{\sigma}, \hat{\boldsymbol{\sigma}}) \equiv \frac{2I(\boldsymbol{\sigma}, \hat{\boldsymbol{\sigma}})}{H(\boldsymbol{\sigma}) + H(\hat{\boldsymbol{\sigma}})}. \tag{9}$$

We adopt this measure as the error function for the readout classifier. For the resulting state $x_{i\mu}^T$, the estimated assignment probability $p_{i\sigma}$ that vertex $i$ belongs to the group $\sigma$ is defined as $p_{i\sigma} \equiv \text{softmax}(a_{i\sigma})$, where $a_{i\sigma} = \sum_\mu x_{i\mu} W_{\mu\sigma}^{\text{out}}$. Each element of the NMI is then obtained as

$$P_{\sigma\hat{\sigma}} = \frac{1}{N} \sum_{i=1}^{N} P(i \in V_\sigma, i \in V_{\hat{\sigma}}) = \frac{1}{N} \sum_{i \in V_\sigma} p_{i\hat{\sigma}},$$

$$P_\sigma = \sum_{\hat{\sigma}} P_{\sigma\hat{\sigma}} = \gamma_\sigma, \qquad P_{\hat{\sigma}} = \sum_\sigma P_{\sigma\hat{\sigma}} = \frac{1}{N} \sum_{i=1}^{N} p_{i\hat{\sigma}}. \tag{10}$$

In summary,

$$\text{NMI}\left([P_{\sigma\hat{\sigma}}]\right) = 2 \left( 1 - \frac{\sum_{\sigma\hat{\sigma}} P_{\sigma\hat{\sigma}} \log P_{\sigma\hat{\sigma}}}{\sum_\sigma \gamma_\sigma \log \gamma_\sigma + \sum_{\sigma\hat{\sigma}} P_{\sigma\hat{\sigma}} \log \sum_\sigma P_{\sigma\hat{\sigma}}} \right). \tag{11}$$

This measure is permutation invariant, because the NMI counts the label co-occurrence patterns for each vertex in $\boldsymbol{\sigma}$ and $\hat{\boldsymbol{\sigma}}$.

## 5   Experiments

First, the consistency between our mean-field theory and a specific implementation of an untrained GNN is examined. The performance of the untrained GNN is evaluated by drawing phase diagrams. For the SBMs with various values for the average degree $c$ and the strength of group structure $\epsilon$, the overlap, i.e., the fraction of vertices that coincide with their planted labels, is calculated. Afterward, it is investigated whether a significant improvement is achieved through the parameter learning of the model. Note that because even a completely random clustering can correctly infer half of the labels on average, the minimum of the overlap is $0.5$.[5] As mentioned above, we adopt $\phi = \tanh$ as the specific choice of activation function.

### 5.1   Untrained GNN with the k-means classifier

We evaluate the performance of the untrained GNN in which the resulting state $\boldsymbol{X}$ is read out using the k-means (more precisely k-means++ [30]) classifier. In this case, no parameter learning takes place. We set the dimension of the feature space to $D = 100$ and the number of layers to $T = 100$, and each result represents the average over $30$ samples.

Figure 3a presents the corresponding phase diagram. The overlap is indicated by colors, and the solid line represents the detectability limit estimated by Eq. (7). The dashed line represents the mean-field estimate of the detectability limit of the spectral method[6] [33, 34], and the shaded area

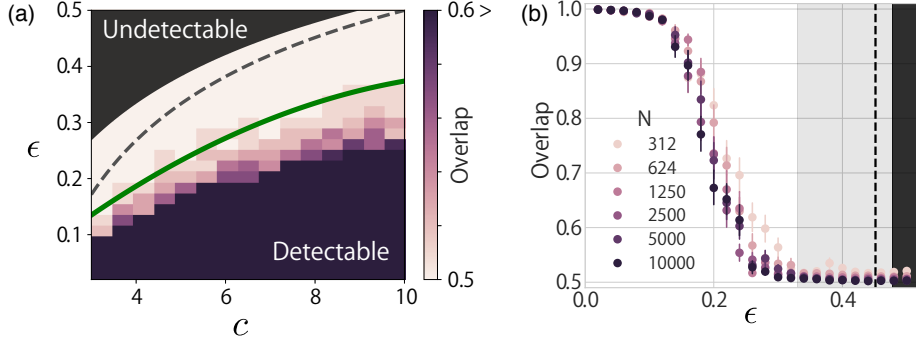

Figure 3: Performance of the untrained GNN using the k-means classifier. (a) The same detectability phase diagram as in Fig. 2. The heatmap represents the overlap obtained using the untrained GNN. (b) The overlaps of the SBM with $c = 8$: The light shaded area represents the region above the estimate using Eq. (7), the dashed line represents the detectability limit of the spectral method, and the dark shaded region represents the information-theoretically undetectable region.

represents the region above which the inference is information-theoretically impossible. It is known that the detectability limit of the BP algorithm [13] coincides with this information-theoretic limit so long as the model parameters are correctly learned. For the present model, it is also known that the EM algorithm can indeed learn these parameters [35]. Note that it is natural that a Bayesian method will outperform others as long as a consistent model is used, whereas it may perform poorly if the assumed model is not consistent.

It can be observed that our mean-field estimate exhibits a good agreement with the numerical experiment. For a closer view, the overlaps of multiple graph sizes with $c = 8$ are presented in Fig. 3b. For $c = 8$, the estimate is $\epsilon^* \approx 0.33$, and this appears to coincide with the point at which the overlap is almost $0.5$. It should be noted that the performance can vary depending on the implementation details. For example, while the k-means method is performed to $\boldsymbol{X}^T$ in the present experiment, it can instead be performed to $\phi\left(\boldsymbol{X}^T\right)$. An experiment concerning such a case is presented in the supplemental material.

## 5.2  GNN with backpropagation and a trained classifier

Now, we consider a trained GNN, and compare its performance with the untrained one. A set of SBM instances is provided as the training set. This consists of $1,000$ SBM instances with $N = 5,000$, where an average degree $c \in \{3, 4, \ldots, 10\}$ and strength of the group structure $\epsilon \in \{0.05, 0.1, \ldots, 0.5\}$ are adopted. For the validation (development) set, $100$ graph instances of the same SBMs are provided. Finally, the SBMs with various values of $\epsilon$ and the average degree $c = 8$ are provided as the test set.

We evaluated the performance of a GNN trained by backpropagation. We implemented the GNN using Chainer (version 3.2.0) [36]. As in the previous section, the dimension of the feature space is set to $D = 100$, and various numbers of layers are examined. For the error function of the readout classifier, we adopted the NMI error function described in Section 4. The model parameters are optimized using the default setting of the Adam optimizer [37] in Chainer. Although we examined various optimization procedures for fine-tuning, the improvement was hardly observable.

We also employ residual networks (ResNets) [38] and batch normalization (BN) [39]. These are also adopted in [32]. The ResNet imposes skip (or shortcut) connections on a deep network, i.e., $x_{i\mu}^{t+1} = \sum_{j\nu} A_{ij}\phi\left(x_{j\nu}^t\right) W_{\nu\mu}^t + x_{i\mu}^{t-s}$, where $s$ is the number of layers skipped, and is set as $s = 5$. The BN layer, which standardizes the distribution of the state $\boldsymbol{X}^t$, is placed at each intermediate layer $t$. Finally, we note that the parameters of deep GNNs (e.g., $T > 25$) cannot be learned correctly without using the ResNet and BN techniques.

The results using the GNN trained as above are illustrated in Fig. 4. First, it can be observed from Fig. 4a that a deep structure is important for a better accuracy. For sufficiently deep networks, the

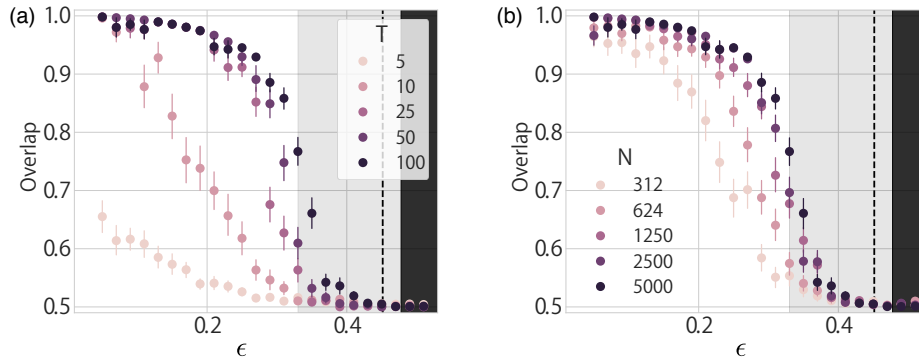

Figure 4: Overlaps of the GNN with trained model parameters. (a) The overlaps of the GNN with various number of layers $T$ on the SBM with $c = 8$ and $N = 5,000$. (b) The graph size dependence of the overlap of the GNN with $T = 100$ on the SBM with $c = 8$. In both cases, the shaded regions and dashed line are plotted in the same manner as in Fig. 3b.

overlaps obtained by the trained GNN are clearly better than those of the untrained counterpart (see Fig. 3b). On the other hand, the region of $\epsilon$ where the overlap suddenly deteriorates still coincides with our mean-field estimate for the untrained GNN. This implies that in the limit $N \to \infty$, the detectability limit is not significantly improved by training. To demonstrate the finite-size effect in the result of Fig. 4a, the overlaps of various graph sizes are plotted in Fig. 4b. The variation of overlaps becomes steeper around $\epsilon^* \approx 0.33$ as the graph size is increased, implying the presence of detectability phase transition around the value of $\epsilon$ predicted by our mean-field estimate.

The untrained and trained GNNs exhibit a clear difference in overlap when $\boldsymbol{X}^T$ is employed as the readout classifier. However, it should be noted that the untrained GNN where $\phi(\boldsymbol{X}^T)$ is adopted as the readout classifier exhibits a performance close to that of the trained GNN. The reader should also bear in mind that the computational cost required for training is not negligible.

## 6 Discussion

In a minimal GNN model, the adjacency matrix $\boldsymbol{A}$ is employed for the connections between intermediate layers. In fact, there have been many attempts [5, 32, 40, 41] to adopt a more complex architecture rather than $\boldsymbol{A}$. Furthermore, other types of applications of deep neural networks to graph partitioning or related problems have been described [6, 7, 42]. The number of GNN varieties can be arbitrarily extended by modifying the architecture and learning algorithm. Again, it is important to clarify which elements are essential for the performance.

The present study offers a baseline answer to this question. Our mean-field theory and numerical experiment using the k-means readout classifier clarify that an untrained GNN with a simple architecture already performs well. It is worth noting that our mean-field theory yields an accurate estimate of the detectability limit in a compact form. The learning of the model parameters by backpropagation does contribute to an improved accuracy, although this appears to be quantitatively insignificant. Importantly, the detectability limit appears to remain (almost) the same.

The minimal GNN that we considered in this paper is not the state of the art for the inference of the symmetric SBM. However, as described in Section 2, an advantage of the GNN is its flexibility, in that the model can be learned in a data-driven manner. For a more complicated example, such as the graphs of chemical compounds in which each vertex has attributes, the GNN is expected to generically outperforms other approaches. In such a case, the performance may be significantly improved thanks to backpropagation. This would constitute an interesting direction for future work. In addition, the adequacy of the NMI error function that we introduced for the readout classifier should be examined in detail.

**Acknowledgments**

The authors are grateful to Ryo Karakida for helpful comments. This work was supported by the New Energy and Industrial Technology Development Organization (NEDO) (T.K. and M. T.) and JSPS KAKENHI No. 18K11463 (T. O.).

## Footnotes

[1]The bias term is only included in this section, and will be omitted in later sections.

[2]Alternatively, it can be regarded that $\mathrm{diag}(b_{i1}^t, \ldots, b_{iD}^t)$ is added to $\boldsymbol{W}^t$ when $\boldsymbol{b}^t$ has common rows.

[3]A matrix $\mathrm{const.}\boldsymbol{I}$ is sometimes added in order to shift the eigenvalues.

[4]Precisely speaking, this is the BP algorithm in which the stochastic block model (SBM) is assumed as the generative model. The SBM is explained below.

[5]For this reason, the overlap is sometimes standardized such that the minimum equals zero.

[6]Again, there are several choices for the matrix to be adopted in the spectral method. However, the Laplacians and modularity matrix, for example, have the same detectability limit when the graph is regular or the average degree is sufficiently large.

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
