[Supplementary Material · sbmgnn_ supplementary.pdf]

# Supplemental Material:
# Mean-field theory of graph neural networks
# in graph partitioning

**Tatsuro Kawamoto, Masashi Tsubaki**
Artificial Intelligence Research Center,
National Institute of Advanced Industrial Science and Technology,
2-3-26 Aomi, Koto-ku, Tokyo, Japan
{kawamoto.tsatsuro, tsubaki.masashi}@aist.go.jp

**Tomoyuki Obuchi**
Department of Mathematical and Computing Science, Tokyo Institute of Technology,
2-12-1 Ookayama Meguro-ku Tokyo, Japan
obuchi@c.titech.ac.jp

## A   Derivation of the self-consistent equation

In this section, the detailed derivation of the self-consistent equation of the covariance matrix is derived. Here, we recast our starting-point equation:

$$1 = \int \mathcal{D}\hat{\mathbf{x}}^{t+1}\mathcal{D}\mathbf{x}^{t+1}\mathrm{e}^{-\mathcal{L}_0}\left\langle \mathrm{e}^{\mathcal{L}_1}\right\rangle_{A,W^t,X^t}, \qquad \begin{cases} \mathcal{L}_0 = \sum_{\sigma\mu}\gamma_\sigma \hat{\mathsf{x}}^{t+1}_{\sigma\mu}\mathsf{x}^{t+1}_{\sigma\mu} \\ \mathcal{L}_1 = \frac{1}{N}\sum_{\mu\nu}\sum_{ij}A_{ij}W^t_{\nu\mu}\hat{\mathsf{x}}^{t+1}_{\sigma_i\mu}\phi(x^t_{j\nu}). \end{cases} \quad (1)$$

### A.1   Random averages over $W^t$ and $A$

We first take the average of $\exp(\mathcal{L}_1)$ over $\boldsymbol{W}^t$. The Gaussian integral with respect to $\boldsymbol{W}^t$ yields

$$\left\langle \mathrm{e}^{\mathcal{L}_1}\right\rangle_{W^t} = \exp\left(\frac{1}{2DN^2}\sum_{\mu\nu}\sum_{ijk\ell}A_{ij}\hat{\mathsf{x}}^{t+1}_{\sigma_i\mu}\phi(x^t_{j\nu})A_{k\ell}\hat{\mathsf{x}}^{t+1}_{\sigma_k\mu}\phi(x^t_{\ell\nu})\right)$$

$$= \exp\left(\frac{1}{2D}\sum_{\sigma_1\sigma_2\sigma_3\sigma_4}\sum_{\mu}\hat{\mathsf{x}}^{t+1}_{\sigma_1\mu}\hat{\mathsf{x}}^{t+1}_{\sigma_3\mu}\sum_{\nu}\psi^{t,\nu}_{\sigma_1\sigma_2}\psi^{t,\nu}_{\sigma_3\sigma_4}\right) = \exp\left(\frac{D}{2}\sum_{\sigma_1\sigma_3}u^{t+1}_{\sigma_1\sigma_3}v^t_{\sigma_1\sigma_3}\right), \quad (2)$$

where we introduce the following quantities:

$$\psi^{t,\nu}_{\sigma\sigma'} \equiv \frac{1}{N}\sum_{i\in V_\sigma}\sum_{j\in V_{\sigma'}}A_{ij}\phi(x^t_{j\nu}), \qquad (3)$$

$$u^{t+1}_{\sigma\sigma'} \equiv \frac{1}{D}\sum_{\mu}\hat{\mathsf{x}}^{t+1}_{\sigma\mu}\hat{\mathsf{x}}^{t+1}_{\sigma'\mu}, \qquad (4)$$

$$v^t_{\sigma\sigma'} \equiv \frac{1}{D}\sum_{\mu}\sum_{\tilde{\sigma}\tilde{\sigma}'}\psi^{t,\mu}_{\sigma\tilde{\sigma}}\psi^{t,\mu}_{\sigma'\tilde{\sigma}'}. \qquad (5)$$

Therefore, Eq. (1) can be written as

$$
1 = \int \mathcal{D}\hat{\mathbf{x}}^{t+1}\mathcal{D}\mathbf{x}^{t+1} \left\langle \left\langle \int \mathcal{D}\hat{\boldsymbol{u}}^{t+1}\mathcal{D}\boldsymbol{u}^{t+1} \int \mathcal{D}\hat{\boldsymbol{v}}^{t}\mathcal{D}\boldsymbol{v}^{t} \int \mathcal{D}\hat{\boldsymbol{\psi}}^{t}\mathcal{D}\boldsymbol{\psi}^{t} \right. \right.
$$

$$
\times \left\langle \exp\left[ -\mathcal{L}_0 + \frac{D}{2}\sum_{\sigma\sigma'} u^{t+1}_{\sigma\sigma'} v^{t}_{\sigma\sigma'} - \sum_{\sigma\sigma'} \hat{u}^{t+1}_{\sigma\sigma'}\left( Du^{t+1}_{\sigma\sigma'} - \sum_{\mu}\hat{\mathsf{x}}^{t+1}_{\sigma\mu}\hat{\mathsf{x}}^{t+1}_{\sigma'\mu} \right) \right. \right.
$$

$$
- \sum_{\sigma\sigma'} \hat{v}^{t}_{\sigma\sigma'}\left( Dv^{t}_{\sigma\sigma'} - \sum_{\nu}\sum_{\tilde{\sigma}\tilde{\sigma}'} \psi^{t,\nu}_{\sigma\tilde{\sigma}}\psi^{t,\nu}_{\sigma'\tilde{\sigma}'} \right)
$$

$$
\left. \left. \left. \left. - \sum_{\sigma\sigma'}\sum_{\mu} \hat{\psi}^{t,\mu}_{\sigma\sigma'}\left( \psi^{t,\mu}_{\sigma\sigma'} - \frac{1}{N}\sum_{i\in V_\sigma}\sum_{j\in V_{\sigma'}} A_{ij}\phi(x^{t}_{j\nu}) \right) \right] \right\rangle_A \right\rangle_{X^t} . \right. \tag{6}
$$

Note that as we will see below, $\boldsymbol{u}^{t+1}$, $\boldsymbol{v}^{t}$, $\boldsymbol{\psi}^{t}$, and their conjugates are related to $X^t$, and thus the average over $X^t$ is taken outside of their integral.

We next take the average over a random graph. In Eq. (6), only the final term in the exponent is relevant to $A$. We denote this term as $\mathcal{L}_2$. We also let $\Xi^{t}_{ij} = \sum_\mu \left( \hat{\psi}^{t,\mu}_{\sigma_i\sigma_j}\phi(x^{t}_{j\mu}) + \hat{\psi}^{t,\mu}_{\sigma_j\sigma_i}\phi(x^{t}_{i\mu}) \right)/N$. Because the graph is generated from the SBM, we have that

$$
\left\langle e^{\mathcal{L}_2} \right\rangle_A = \prod_{i<j}\left[ \sum_{A_{ij}\in\{0,1\}} \left( \rho_{\sigma_i\sigma_j}e^{\Xi^{t}_{ij}} \right)^{A_{ij}} \left( 1-\rho_{\sigma_i\sigma_j} \right)^{1-A_{ij}} \right] \tag{7}
$$

$$
\approx \exp\left[ \sum_{i<j}\rho_{\sigma_i\sigma_j}\left( e^{\Xi^{t}_{ij}} - 1 \right) \right] \tag{8}
$$

$$
\approx \exp\left[ \sum_{i<j}\rho_{\sigma_i\sigma_j}\Xi^{t}_{ij} \right] \tag{9}
$$

$$
= \exp\left[ \sum_\mu\sum_{\sigma\sigma'}\gamma_\sigma\rho_{\sigma\sigma'}\hat{\psi}^{t,\mu}_{\sigma\sigma'}\sum_{j\in V_{\sigma'}}\phi(x^{t}_{j\mu}) \right] . \tag{10}
$$

At the second line, we used the fact that $\rho_{\sigma\sigma'} = O(N^{-1})$. Then, at the third line we used $\Xi^{t}_{ij} = O(D/N) \ll 1$. Finally, at the last line we used the symmetry of the undirected graph, $\rho_{\sigma\sigma'} = \rho_{\sigma'\sigma}$.

Note here that the degrees of freedom with respect to the feature dimension are factored out, and thus the dependence on $\mu$ can be omitted. Hereafter, the same notation will be employed for the variables without the $\mu$-dependence. We also introduce the notation $\exp^D(f) \equiv \exp(Df)$. The factor inside of the average over $X^t$ in Eq. (6) can be written as follows:

$$
\int \mathcal{D}\hat{\boldsymbol{u}}^{t+1}\mathcal{D}\boldsymbol{u}^{t+1} \int \mathcal{D}\hat{\boldsymbol{v}}^{t}\mathcal{D}\boldsymbol{v}^{t} \int \mathcal{D}\hat{\boldsymbol{\psi}}^{t}\mathcal{D}\boldsymbol{\psi}^{t}\, e^{\mathcal{L}^*(\hat{\boldsymbol{u}}^{t+1})} \times \exp^D\left[ \sum_{\sigma\sigma'} B_{\sigma\sigma'}\hat{\psi}^{t}_{\sigma\sigma'}\frac{1}{\gamma_{\sigma'}N}\sum_{j\in V_{\sigma'}}\phi(x^{t}_j) \right.
$$

$$
\left. + \sum_{\sigma\sigma'}\left( \frac{1}{2}u^{t+1}_{\sigma\sigma'}v^{t}_{\sigma\sigma'} - \hat{u}^{t+1}_{\sigma\sigma'}u^{t+1}_{\sigma\sigma'} - \hat{v}^{t}_{\sigma\sigma'}v^{t}_{\sigma\sigma'} - \hat{\psi}^{t}_{\sigma\sigma'}\psi^{t}_{\sigma\sigma'} - \hat{v}^{t}_{\sigma\sigma'}\sum_{\tilde{\sigma}\tilde{\sigma}'}\psi^{t}_{\sigma\tilde{\sigma}}\psi^{t}_{\sigma'\tilde{\sigma}'} \right) \right] , \tag{11}
$$

where

$$
\mathcal{L}^*(\hat{\boldsymbol{u}}^{t+1}) = \log\int \mathcal{D}\hat{\mathbf{x}}^{t+1}\mathcal{D}\mathbf{x}^{t+1}\exp^D\left( -\sum_\sigma\gamma_\sigma\hat{\mathsf{x}}^{t+1}_\sigma\mathsf{x}^{t+1}_\sigma + \sum_{\sigma\sigma'}\hat{u}^{t+1}_{\sigma\sigma'}\hat{\mathsf{x}}^{t+1}_\sigma\hat{\mathsf{x}}^{t+1}_{\sigma'} \right) . \tag{12}
$$

As in the main text, we have defined $B_{\sigma\sigma'} \equiv N\gamma_\sigma\rho_{\sigma\sigma'}\gamma_{\sigma'}$.

When $D \gg 1$, the saddle-point condition of the exponent in Eq. (11) yields $u^{t+1}_{\sigma\sigma'} = \left\langle \hat{\mathsf{x}}^{t+1}_\sigma\hat{\mathsf{x}}^{t+1}_{\sigma'} \right\rangle_{\mathcal{L}^*}$, $v^{t}_{\sigma\sigma'} = \psi^{t}_\sigma\psi^{t}_{\sigma'}$, $\psi^{t}_{\sigma\sigma'} = B_{\sigma\sigma'}(\gamma_{\sigma'}N)^{-1}\sum_{j\in V_{\sigma'}}\phi(x^{t}_j)$, $\hat{u}^{t+1}_{\sigma\sigma'} = v^{t}_{\sigma\sigma'}/2$, $\hat{v}^{t}_{\sigma\sigma'} = u^{t+1}_{\sigma\sigma'}/2$, and $\hat{\psi}^{t}_{\sigma\sigma'} =$

$\sum_{\tilde{\sigma}} (\hat{v}_{\sigma\tilde{\sigma}}^t + \hat{v}_{\tilde{\sigma}\sigma}^t) \psi_{\tilde{\sigma}}^t$, where $\psi_\sigma^t \equiv \sum_{\tilde{\sigma}} \psi_{\sigma\tilde{\sigma}}^t$, and $\langle \ldots \rangle_{\mathcal{L}^*}$ is the average taken with the weight of the integrand of Eq. (12). Because the correlation between the auxiliary variables should be zero, owing to causality [1, 2], we finally arrive at

$$1 = \int \mathcal{D}\hat{\mathbf{x}}^{t+1} \mathcal{D}\mathbf{x}^{t+1} \exp\left(-\sum_\sigma \gamma_\sigma \hat{x}_\sigma^{t+1} x_\sigma^{t+1}\right) \left\langle \exp\left(\frac{1}{2} \sum_{\sigma\sigma'} \hat{x}_\sigma^{t+1} F_{\sigma\sigma'}\left(\boldsymbol{X}^t\right) \hat{x}_{\sigma'}^{t+1}\right) \right\rangle_{X^t}, \quad (13)$$

where $F_{\sigma\sigma'}\left(\boldsymbol{X}^t\right)$ is defined as

$$F_{\sigma\sigma'}\left(\boldsymbol{X}^t\right) \equiv \sum_{\tilde{\sigma}\tilde{\sigma}'} B_{\sigma\tilde{\sigma}} B_{\sigma'\tilde{\sigma}'} \frac{1}{\gamma_{\tilde{\sigma}} N} \frac{1}{\gamma_{\tilde{\sigma}'} N} \sum_{k\in V_{\tilde{\sigma}}} \sum_{\ell\in V_{\tilde{\sigma}'}} \phi(x_k^t)\phi(x_\ell^t). \quad (14)$$

## A.2  Stochastic process with a correlated noise

Here, we compare Eq. (13) with a Markovian discrete-time stochastic process $y_\sigma^{t+1} = \eta_\sigma^t$, in which each element is correlated via a random noise, i.e., $\langle \eta_\sigma^t \rangle_\eta = 0$, and $\langle \eta_\sigma^t \eta_{\sigma'}^t \rangle_\eta = C_{\sigma\sigma'}$ for any $t$. The corresponding normalization condition reads

$$1 = \int \prod_\sigma dy_\sigma^{t+1} \left\langle \prod_\sigma \delta\left(y_\sigma^{t+1} - \eta_\sigma^t\right) \right\rangle_\eta = \int \prod_\sigma \gamma_\sigma \frac{d\hat{y}_\sigma^{t+1} dy_\sigma^{t+1}}{2\pi i} \left\langle e^{-\sum_\sigma \gamma_\sigma \hat{y}_\sigma^{t+1}\left(y_\sigma^{t+1} - \eta_\sigma^t\right)} \right\rangle_\eta$$

$$= \int \mathcal{D}\hat{\boldsymbol{y}}^{t+1} \mathcal{D}\boldsymbol{y}^{t+1} e^{-\sum_\sigma \gamma_\sigma \hat{y}_\sigma^{t+1} y_\sigma^{t+1}} \int \prod_\sigma \frac{d\eta_\sigma^t}{2\pi i} \exp\left(-\frac{1}{2} \sum_{\sigma\sigma'} \eta_\sigma^t C_{\sigma\sigma'}^{-1} \eta_{\sigma'}^t + \sum_\sigma \gamma_\sigma \hat{y}_\sigma^{t+1} \eta_\sigma^t\right)$$

$$= \int \mathcal{D}\hat{\boldsymbol{y}}^{t+1} \mathcal{D}\boldsymbol{y}^{t+1} \exp\left(-\sum_\sigma \gamma_\sigma \hat{y}_\sigma^{t+1} y_\sigma^{t+1} + \frac{1}{2} \sum_{\sigma\sigma'} \hat{y}_\sigma^{t+1} \gamma_\sigma C_{\sigma\sigma'} \gamma_{\sigma'} \hat{y}_{\sigma'}^{t+1}\right). \quad (15)$$

Analogously to the case of the GNN, we have defined $\mathcal{D}\hat{\boldsymbol{y}}^{t+1} \mathcal{D}\boldsymbol{y}^{t+1} \equiv \prod_\sigma \gamma_\sigma d\hat{y}_\sigma^{t+1} dy_\sigma^{t+1}/2\pi i$.

## A.3  Self-consistent equation

Finally, we compare Eqs. (13) and (15). However, note that these are not of exactly the same form, because the average over $\boldsymbol{X}^t$ is taken outside of the exponential in Eq. (13). Two approximations are made in order to derive the self-consistent equation, and the assumptions that justify these approximations are discussed afterward.

First, if the approximation

$$\left\langle \exp\left(\frac{1}{2} \sum_{\sigma\sigma'} \hat{x}_\sigma^{t+1} F_{\sigma\sigma'}\left(\boldsymbol{X}^t\right) \hat{x}_{\sigma'}^{t+1}\right) \right\rangle_{X^t} \approx \exp\left(\frac{1}{2} \sum_{\sigma\sigma'} \hat{x}_\sigma^{t+1} \left\langle F_{\sigma\sigma'}\left(\boldsymbol{X}^t\right) \right\rangle_{X^t} \hat{x}_{\sigma'}^{t+1}\right) \quad (16)$$

holds in the stationary limit, then the group-wise state $x^t$ can be regarded as a Gaussian variable whose correlation matrix obeys

$$C_{\sigma\sigma'} = \frac{1}{\gamma_\sigma \gamma_{\sigma'}} \sum_{\sigma_1 \sigma_2} B_{\sigma\tilde{\sigma}} B_{\sigma'\tilde{\sigma}'} \left\langle \frac{1}{\gamma_{\tilde{\sigma}} N \gamma_{\tilde{\sigma}'} N} \sum_{i\in V_{\tilde{\sigma}}} \sum_{j\in V_{\tilde{\sigma}'}} \phi(x_i)\phi(x_j) \right\rangle_{X^t}. \quad (17)$$

This equation is still not closed, because the right-hand side of Eq. (17) depends on the statistic of $\boldsymbol{X}^t$, rather than $x^t$. However, because the vertices within the group $\sigma$ are statistically equivalent, $\{x_i\}_{i\in V_\sigma}$ are expected to obey the same distribution with mean $x_\sigma$, which itself is a random variable. If $\sum_{i\in V_\sigma} \phi(x_i)/(\gamma_\sigma N) \approx \phi(x_\sigma)$ holds, then the right-hand side of Eq. (17) can be evaluated as the average with respect to the group-wise variable $x^t$. Then, within this regime we arrive at the following self-consistent equation with respect to the covariance matrix $\boldsymbol{C} = [C_{\sigma\sigma'}]$:

$$C_{\sigma\sigma'} = \frac{1}{\gamma_\sigma \gamma_{\sigma'}} \sum_{\tilde{\sigma}\tilde{\sigma}'} B_{\sigma\tilde{\sigma}} B_{\sigma'\tilde{\sigma}'} \int \frac{d\mathbf{x}\, e^{-\frac{1}{2}\mathbf{x}^\top \boldsymbol{C}^{-1} \mathbf{x}}}{(2\pi)^{\frac{N}{2}} \sqrt{\det \boldsymbol{C}}} \phi(x_{\tilde{\sigma}})\phi(x_{\tilde{\sigma}'}). \quad (18)$$

Let us consider the first approximation that we adopted in Eq. (16). In the terminology of physics, this is the replacement of a free energy with an internal energy, or the neglect of the entropic contribution. It is difficult to evaluate this residual in general. However, note that this becomes closer to equality as every $x_i$ approaches the same value. Therefore, this implies that the self-consistent equation is more accurate as we approach the detectability limit, and yields an adequate estimate of the critical value.

Let us next consider the second approximation we adopted in Eq. (17). Although the law of large numbers with respect to $\phi(x_i)$ (not $x_i$) ensures that $\sum_{i \in V_\sigma} \phi(x_i)/(\gamma_\sigma N)$ has a certain value characterized by the group, this may be different from $\phi(\mathsf{x}_\sigma)$. In fact, the relation between these is in general an inequality (Jensen's inequality) when the activation function $\phi$ is a convex function. The (exact) equality holds only when $\{x_i\}$ is constant or the function $\phi$ is linear within the considered domain.

The second approximation can be justified in the following cases. The first case is when the fluctuation of $x_i - \mathsf{x}_{\sigma_i}$ is negligible compared to the magnitude of $\mathsf{x}_\sigma$. Note that this is the same assumption as we made in the first approximation. To see this precisely, let us express $x_i$ as $x_i = \mathsf{x}_\sigma + z_i$ for $i \in V_\sigma$. We can formally write the probability distribution $P(\{x_i\})$ of $\{x_i\}$ in a hierarchical fashion as follows:

$$P(\{x_i\}) = \int \prod_\sigma d\mathsf{x}_\sigma \int \prod_i dz_i P_{\boldsymbol{\sigma}}(\{\mathsf{x}_\sigma\}) P(\{x_i\}) \prod_i \delta(z_i - x_i + \mathsf{x}_{\sigma_i}), \qquad (19)$$

where $P_{\boldsymbol{\sigma}}(\{\mathsf{x}_\sigma\})$ is the probability distribution with respect to $\mathsf{x}$. Thus, the expectation $\langle f(\boldsymbol{X}) \rangle_{\boldsymbol{X}}$ can be expressed as

$$\langle f(\boldsymbol{X}) \rangle_{\boldsymbol{X}} \equiv \int \prod_i dx_i P(\{x_i\}) f(\{x_i\})$$

$$= \int \prod_\sigma d\mathsf{x}_\sigma \int \prod_i dz_i P(\{z_i\}|\{\mathsf{x}_\sigma\}) P_{\boldsymbol{\sigma}}(\{\mathsf{x}_\sigma\}) f(\{\mathsf{x}_{\sigma_i} + z_i\}), \qquad (20)$$

where $P(\{z_i\}|\{\mathsf{x}_\sigma\}) \equiv P(\{\mathsf{x}_{\sigma_i} + z_i\})$, which can be a nontrivial function. However, whenever the contributions from the average with respect to $z_i$ are negligible, Eq. (20) implies that the expectation in Eq. (17) can be evaluated using only the group-wise variables $\{\mathsf{x}_\sigma\}$. Another case is when the activation function $\phi$ is almost linear within the domain over which $z_i$ fluctuates. For example, in the case that $\phi = \tanh$, the present approximation does not deteriorate the accuracy even when $\mathsf{x}_\sigma \approx 0$. When either of these assumption holds, the equality of Jensen's inequality is approximately satisfied, and our derivation of the self-consistent equation is justified.

## B  K-means classification using $\phi(\boldsymbol{X})$

Instead of $\boldsymbol{X}^T$, $\phi(\boldsymbol{X}^T)$ can be adopted to perform the k-means classification after the feedforward process. Again, we employ $\tanh$ as the nonlinear activation function. The results of an untrained GNN and a trained GNN under the same experimental settings as in the main text are illustrated in Fig. 1 and Fig. 2, respectively. In Fig. 1a, the reader should note that the range of the color gradient is different from that in the phase diagram in the main text. For the untrained GNN, the obtained overlaps are clearly better than that using $\boldsymbol{X}^T$. It can be understood that the error is reduced because the nonlinear function drives each element of the state $\boldsymbol{X}^T$ to either $+1$ or $-1$, making the classification using the k-means method easier and more accurate. On the other hand, for the trained GNN, differences between the overlaps using $\boldsymbol{X}^T$ and $\phi(\boldsymbol{X}^T)$ are hardly observable.

Particularly for the case of an untrained GNN in which $\phi(\boldsymbol{X}^T)$ is adopted for the readout classifier, the overlap gradually changes around the estimated detectability limit. This may be as result of the strong finite-size effect. Again, note that our estimate of the detectability limit is for the case that $N \to \infty$.