[Reviews · NeurIPS 2018]

Reviewer 1



A GNN (graph neural network) is a neural network whose input is a graph. This paper studies the problem of using a GNN to detect clusters in a graph drawn from the 2-groups SBM (stochastic block model: a popular model for random graphs with community structure). Although it is already known how to optimally solve the SBM (i.e. find the hidden clusters), the GNN has the advantage of being a data-driven approach that does not rely on knowing the specific random model. Thus, it is of interest to analyze the performance of GNN on the SBM. In the GNN architecture studied here, each layer has a node for each vertex in the graph. A node in the GNN is connected to its neighbors (according to the graph adjacency matrix) in the previous layer. Each node of the GNN stores a D-dimensional feature vector. This paper analyzes the performance of an untrained GNN on the SBM. This means the GNN weights are chosen at random, the GNN runs in order to produce a D-dimensional feature vector for each vertex, and then these vectors are clustered via k-means in order to partition the vertices. The analysis is not rigorous but is a heuristic calculation based on the mean-field approximation from statistical physics. The derivation yields a conjectured threshold (depending on the SBM parameters) at which the GNN is able to detect the presence of clusters and obtain a partition that non-trivially correlates with the truth. The authors include numerical experiments, showing that the analytically-derived threshold appears to agree reasonably well with the experimental performance of the untrained GNN. They also experimentally compare the untrained GNN with a trained one. They see that the training does not seem to significantly affect the detection threshold (at which point non-trivial recovery becomes possible) but training does significantly improve reconstruction accuracy once you're above the threshold. I think that this paper constitutes a decent start to the rigorous study of GNN's but I'm unsure whether its contributions are enough to merit acceptance to NIPS. The results are somewhat modest because they only analyze the untrained GNN and the analysis is non-rigorous. Still, the result obtained seems to reasonably approximate the truth, and the experimental results of the paper are of value. I suspect that the derivation in this paper may not quite give the correct result. To get a rigorously-correct (in the large-graph limit) analysis of the constant-degree SBM typically requires a self-consistency equation simliar to (7) whose state is an entire distribution (see e.g. https://arxiv.org/abs/1611.00814), whereas this paper seems to only keep track of the mean value and make a Gaussian approximation. Regardless, the computations in this paper are at least a crude approximation to the correct result. EDIT: After reading the other reviews and the author feedback, I have changed my overall score from 5 to 6. The authors make a fair point that deriving a crude approximation to the threshold still has value. I still think it would be good to derive the exact expression for the threshold (in the large graph limit). I would suggest that the authors address whether they believe their derivation gives the exact threshold (it sounds like the answer is no?) and whether deriving the exact threshold (even non-rigorously) is within reach of established techniques from statistical physics (I suspect the answer should be yes). In any case, I have no objections to accepting the paper in its current form.

Reviewer 2



The current advances in neural networks have amazing experimental results but they lack good theoretical understanding. This paper is another attempt in linking some neural networks to methods of statistical physics (see e.g. also “The Loss Surfaces of Multilayer Networks”, Choromanska et al., AISTATS 2015). The paper has several very interesting ideas, derivations, and insights, even though the final results are not overwhelming.

Reviewer 3



In this paper, the authors develop a mean-field theory for a model graph neural network and they show that it accurately predicts performance of random GNNs assigning labels to the stochastic block model. Overall, I found this paper to be a good submission that provides insight into an increasingly popular family of models. The theory seemed correct and the experiments seemed carefully done. However, I do think the paper would benefit from clarity in the overall presentation. Some specific feedback: 1) I think the paper might benefit from a related works section, especially discussing how the methods here relate to other papers that have used the mean field approach. 2) Some notation is not defined. For example, a) The c parameter in equation 4 is not defined until much later in the text. b) The node state matrix at t=0. 3) I would have loved to see some discussion of GNNs using performance on the model task that could inform more general work on graph neural networks. For example, if I use weights whose variance is \sigma^2 / N is there any dependence in model performance on \sigma? Does this dependence carry over when using GNNs on other problem areas?